# Polyvinylidene Fluoride/Aromatic Hyperbranched Polyester of Third-Generation-Based Electrospun Nanofiber as a Self-Powered Triboelectric Nanogenerator for Wearable Energy Harvesting and Health Monitoring Applications

**DOI:** 10.3390/polym15102375

**Published:** 2023-05-19

**Authors:** Ramadasu Gunasekhar, Ponnan Sathiyanathan, Mohammad Shamim Reza, Gajula Prasad, Arun Anand Prabu, Hongdoo Kim

**Affiliations:** 1Department of Chemistry, School of Advanced Sciences, Vellore Institute of Technology, Vellore 632014, India; 2Department of Advanced Materials Engineering for Information & Electronics, College of Engineering, Kyung Hee University, Yongin-si 17104, Gyeonggi-do, Republic of Korea; 3School of Energy, Materials and Chemical Engineering, Korea University of Technology and Education, 1600, Cheonan-si 31253, Chungcheongnam-do, Republic of Korea

**Keywords:** electrospinning, triboelectric nanogenerator, PVDF, hyperbranched polyester

## Abstract

Flexible pressure sensors have played an increasingly important role in the Internet of Things and human–machine interaction systems. For a sensor device to be commercially viable, it is essential to fabricate a sensor with higher sensitivity and lower power consumption. Polyvinylidene fluoride (PVDF)-based triboelectric nanogenerators (TENGs) prepared by electrospinning are widely used in self-powered electronics owing to their exceptional voltage generation performance and flexible nature. In the present study, aromatic hyperbranched polyester of the third generation (Ar.HBP-3) was added into PVDF as a filler (0, 10, 20, 30 and 40 wt.% w.r.t. PVDF content) to prepare nanofibers by electrospinning. The triboelectric performances (open-circuit voltage and short-circuit current) of PVDF-Ar.HBP-3/polyurethane (PU)-based TENG shows better performance than a PVDF/PU pair. Among the various wt.% of Ar.HBP-3, a 10 wt.% sample shows maximum output performances of 107 V which is almost 10 times that of neat PVDF (12 V); whereas, the current slightly increases from 0.5 μA to 1.3 μA. The self-powered TENG is also effective in measuring human motion. Overall, we have reported a simpler technique for producing high-performance TENG using morphological alteration of PVDF, which has the potential for use as mechanical energy harvesters and as effective power sources for wearable and portable electronic devices.

## 1. Introduction

Recently, the need for human interaction applications including synthetic e-skin and health management (temperature, blood pressure, pulse rate, respiration and heartbeat) robots has increased due to the development of self-powered adaptable wearables [1,2,3,4,5,6,7,8,9,10,11,12,13,14]. Human-generated, low-frequency mechanical energy (body movements) can be transformed into electrical energy in wearable devices, which can effectively be used as energy harvesters [15,16]. Specifically, flexible pressure sensors have considerable potential for disease diagnostics, since they can transform external force stimuli into electrical impulses to track human biomedical parameters [17,18,19,20]. Most strain sensors are powered by backup batteries, which makes the sensor bulky. In addition, batteries must be recharged or replaced, which does not meet the needs of wearable electronics [7,21].

An example of such a technique is the nanogenerator, which transforms mechanical energy into electricity, and is the final outcome of a minor physical change [6]. They have several virtues, such as being compact, affordable, stable, etc. They are also most suitable for use in self-powered devices [6,18,22,23,24,25,26,27]. Electrostatic induction and contact electrification are the operating mechanisms of the nanogenerator [28]. Sensitive technologies such as those that are self-powered are desired in order to tackle the aforementioned challenges. Initiated by Wang’s group, and depending on the effects of contact electrification and electrostatic induction, triboelectric nanogenerators (TENGs) can transform an implied mechanical force into an electrical signal [29]. TENGs have already been explored extensively as well as being applied in self-powered sensors due to their flexible device structural system, maximum output energy, high energy-conversion efficiency, and wide range of available materials [11,30,31,32,33]. To examine human activity, the TENG strain sensor’s sensitivity should be further increased [6,34,35,36,37].

TENGs can operate in a variety of ways, but the majority of them have been observed to operate in contact separation, which occurs when two surfaces with two electron affinities encounter each other [6,38]. The quantity of the transferred charge resulting from contact electrification is determined by the number of electrons lost or gained on the frictional surface [11]. Thus, it can raise the charges just on triboelectric surfaces to enhance the polarity distinction between the two contacting layers, thereby improving output triboelectric efficiency [39]. Different materials have been tested during the past few years, and a triboelectric series has been suggested [40,41,42]. It has been proven successful to increase charge density in recent attempts to modify the chemical properties of polymer surfaces by adding moieties with various electron affinities [39,43]. Through adding suitable layers on top of the triboelectric layer to increase the polarity difference between the two contacting layers, it provides a distinct method for improving the performance of a self-powered triboelectric sensor [44].

In TENG, there is always a pair of triboelectric materials with different charge affinities that are employed to generate charges. To generate more charges or obtain a higher output from the TENG, a significant difference in the charge affinities of the two materials is preferred. However, in practice, the two materials with the highest difference in charge affinity are not selected. The reason is that the triboelectrification between two materials is based not only on their chemical compositions but also on other physical features such as their elasticity, friction and topographical structure. The triboelectric effect and level of contact electrification performance are based on materials that come into contact. In this case, contact electrification is caused by the surface transfer of electrons or ions between these two materials. Each material has its own ability to lose or gain electrons during the contact electrification process. This ability can be found in a list of materials (triboelectric series), where the polarity and amount of charge for each material are described as shown in Appendix A. For example, a material with fluorine functional groups can attract as many electrons as possible from its counterparts. Polyvinylidene fluoride (PVDF) is a material that tends to gain electrons during electrification. On the other hand, PU is positively charged during electrification. Both materials lie far enough apart in the triboelectric series, and this distance indicates their high triboelectric potential on contact.

Additionally, the technique used for fabrication is crucial in developing high-performance nanogenerators. Due to its adaptability in applications including chemical and physical sensors, healthcare diagnostics, sensitive robotics and wearable technology, fiber-based electronic devices are becoming more and more popular these days [20,29]. In the case of PVDF, it is one of the most studied materials as a nanogenerator; its electrospinning combines mechanical stretching as well as electric poling into a single process for obtaining PVDF in the electroactive *β*-phase. It is claimed that, in comparison with its as-cast film, PVDF nanofibers produced by the electrospinning technique exhibit a greater coefficient of piezoelectricity and higher energy conversion efficiency [5,6,45,46,47,48,49,50,51]. Some of the positive outcomes of obtaining electrospun fibers are higher surface area-to-volume ratio, availability of diversified polymeric materials, combination of the materials’ flexibilities, comparatively lesser manufacturing expenses and ease in depositing fibers on various types of substrates [52,53,54,55,56,57,58].

Electrospun PVDF nanofibers offer a lot of prospects for applications such as energy harvesters [6,54,59,60,61] and sensor applications [17,62,63,64,65]. The selection of triboelectric material in nanogenerator fabrication is as important as the fabrication technique used [5,6,54,66]. To ensure excellent electrical output, several types of materials such as organic, inorganic, polymeric, piezoelectric, triboelectric and dielectric materials were investigated [6,54,67,68,69]. Amongst the most adaptable polymers, PVDF is applied in a wide range of applications including nanogenerators, lithium-ion batteries, semiconductors, medical and healthcare devices [47,48,50,62,70,71,72,73]. Furthermore, when mechanical force is applied, the pristine PVDF-based TENG provides relatively minimal electrical output. Because of the all-trans (TTTT) conformation, it shows five distinct crystal phases (*α*, *β*, *γ*, *δ* and *ε*) [74,75,76]. PVDF’s *β*-crystalline phase has the maximum dielectric and piezoelectric response compared to its other polymorphic phases [72,77]. As a result, PVDF with an improved *β*-phase fraction may provide the required electrical efficiency in nanogenerators. As a polymer with multiple functions, PVDF also has excellent ferroelectric, piezoelectric, pyroelectric, and dielectric characteristics [6,54,68,71]. To generate substantial electrical properties suitable for achieving a commercially viable nanogenerator, higher *β*-phase content in PVDF is required, and this is achieved by mechanical stretching, annealing, electrical polling, and by adding diverse additives such as ZnO, RGO: Fe, ALO/RGO, aromatic hyperbranched polyesters (Ar.HBP), etc. The main criteria are the rearrangements of C–F groups to achieve a favorable enhancement in PVDF’s *β*-phase, which in turn can boost the device conductivity of nanogenerators [70,73,78,79,80,81,82,83,84,85,86,87,88,89,90,91,92,93,94,95,96]. Due to their extremely compatible and less expensive characteristics, electrospun fibers are preferentially used in many health-monitoring applications [6,54].

In this study, we fabricated a TENG using electrospun nanofiber based on PVDF-Ar.HBP of the third generation (P-Ar.HBP-3) blend mixture and non-woven PU film, and used it for harvesting energy under diverse conditions. Electrospun P-Ar.HBP-3 nanofiber and non-woven PU film operate as the corresponding negative and positive triboelectric layers. FTIR, XRD, FESEM and EDS techniques were used to analyze and correlate the structural and morphological characteristics of the blend with its triboelectric behavior. Triboelectric performance of the TENG device was assessed using open-circuit voltage (*V*_OC_) and short circuit current (*I*_SC_) parameters. We have demonstrated that the triboelectric property of PVDF can be significantly enhanced through the addition of Ar.HBP of the third generation (Ar.HBP-3) and aided by electrospinning. Hence, a high-performance P-Ar.HBP-3 (10 wt.%) TENG has been successfully developed based on this negative triboelectric material. Detailed investigation revealed that the modulation of PVDF composition using Ar.HBP-3 and nanofiber structure fabricated through electrospinning significantly contributed to the triboelectric performance enhancement of P-Ar.HBP-3 nanofiber. Draft comparison for the above-mentioned voltage obtained by P-Ar.HBP-3 (10 wt.%) in comparison with earlier reported materials is given in Appendix A. Eventually, TENG was demonstrated to harvest mechanical energy suitable for LED lighting and wearable applications for monitoring healthcare.

## 2. Materials and Methods

### 2.1. Materials

PVDF, dimethyl formamide (DMF) and acetone were purchased from Sigma-Aldrich, Seoul, Republic of Korea. Synthesis and characterization of aromatic hyperbranched polyester of third generation (Ar.HBP-3) is reported in our earlier study [69]. All the chemicals and reagents were utilized without any additional purification or processing.

### 2.2. Preparation of P-Ar.HBP-3 Electrospun Nanofibers

P-Ar.HBP-3 with varying Ar.HBP-3 content (10 to 40 wt.%) was dispersed in DMF/acetone (3:2, *v*/*v*) solvent mixture (16 wt.%) under steady magnetic stirring at 350 rpm speed for 4 to 5 h at 50 °C, and the resultant homogenous solution is further used for the electrospinning process. Along with that, comparative studies were carried out using a neat PVDF solution. Custom-designed electrospinning equipment was used to conduct the electrospinning process. A 10 mL syringe was adapted for filling the prepared solution and transferred through a needle having a diameter of 23 G with a flow rate of 1.2 mL/h by a pumped syringe. The distance between the electrospinning needle tip and the film collector wrapped on a roller was 15 cm, and the electrospinning applied voltage was set at 20 kV. For about 6 h, collection of all the fibers was performed with 60 rpm roller speed. During the electrospinning process, fast evaporation of the solvents takes place. We also took additional care to remove the traces of residual solvents, if any, in the as-spun fibers by keeping the samples in vacuum oven for 5 h at 50 °C to remove any traces of residual solvent. Further, we preserved the samples in a desiccator under closed vacuum for a week before proceeding to electrical measurements.

### 2.3. Nanogenerator Fabrication

For the TENG fabrication, P-Ar.HBP-3 (10 to 40 wt.%) electrospun mat and non-woven polyurethane (PU) film were cut into the specified sizes (effective working area: 2 × 2 cm^2^). Across both sides of the mat, two Ni–Cu conductive electrodes were placed as the top and bottom electrodes. Electrodes were then soldered with Cu and finally, two flexible PET substrates of a larger size compared to the size of the fiber mat were fixed on both sides of the electrodes. To construct a compressed device and resolve the gap among materials, the layered structure was finally subjected to proper pressure. This was to prevent the triboelectric effect and to shield the nanogenerator from environmental damage. In addition, this will not allow direct contact of the sensor with the skin and is safe for the user performing the measurements. Appendix A depicts the device’s optical view.

### 2.4. Characterization

Polymer structure was confirmed using Fourier-transform infrared (FTIR, Thermo Nicolet iS50, Thermo Fisher Scientific, Waltham, MA, USA) spectroscopy. Morphology of the fibers was examined using a field emission-scanning electron microscope (FE-SEM, Thermo Fisher FEI QUANTA 250 FEG). The nanofiber’s X-ray diffraction (XRD) patterns were measured using an X-ray diffractometer (Bruker D8 Advance, Mannheim, Germany) with a Cu Kα radiation source. An external pushing force was generated by an electrical pushing machine. *V*_OC_ and *I*_SC_ electrical output parameters were evaluated using a digital phosphor oscilloscope (DPO4104, Tektronix, Beaverton, OR, USA) with an input impedance of 40 MΩ. Furthermore, an oscilloscope was coupled to a low-noise current preamplifier (SR570, Stanford Research Systems, Stanford, CA, USA) for *I*_SC_ measurement. Sensitivity of the TENG was measured using a BIOPAC System, Inc., Goleta, CA, USA MP150 connected to a measurement expertise Piezo Film Lab Amplifier (conditions: *R*_in_ of 100 MΩ and gain of 0 dB). These measurements were carried out for healthcare monitoring applications.

## 3. Results and Discussion

The schematic representation for the production of electrospun nanofibers and TENG for applications in energy harvesting and health monitoring is shown in Figure 1a. Figure 1a(i,ii) demonstrate the preparation of a precursor polymeric solution. In the beginning, PVDF and Ar.HBP-3 (0 to 40 wt.%) were dissolved in DMF: acetone solvent mixture (3:2 *v*/*v*). Afterwards, the polymeric solution was electrospun onto the substrate (Figure 1a(iii)), peeled off, and further TENG fabrication was performed using the nanoweb. Figure 1a(iv) illustrates the FE-SEM imaging of the produced electrospun fibers. Furthermore, the fabricated triboelectric nanofibers based on P-Ar.HBP-3 and PU, as shown in Figure 1a(v), were cut to the desired size (2 × 2 cm^2^) and adhered along with the Ni–Cu electrode and PET substrate, respectively. These devices could be used in a variety of healthcare-monitoring applications.

### 3.1. Working Principle

Mechanical damage to the sensor setup is a big challenge, preventing TENGs’ practical applications. TENGs have four working modes, i.e., single-electrode, contact-separation, lateral-sliding. and free-standing triboelectric layer modes. For single-electrode and contact-separation modes, TENGs are subjected to constant external tapping, bending, and stretching, which will cause mechanical damage such as cracks, leading to rupture of the sensor structure. In the case of lateral-sliding and free-standing triboelectric modes, TENGs suffer from significant friction, wear, and adhesion of material during sliding. For example, the triboelectric surface of TENGs is prone to being worn out by frictional forces. The worn particles are transferred to the surface of the opposite material, resulting in output degradation and a loss in sensor longevity. Our main focus in this study is the enhancement of the PVDF *β*-phase by blending with Ar.HBP-3 so as to increase the output voltage of the sensor. Hence, we did not perform any mechanical studies for the sensor.

As shown in Figure 1b, we considered P-Ar.HBP-3/PU composition based on a contact-separation mechanism to describe the TENG device working mechanism for the developed sensor. In this case, the P-Ar.HBP-3 nanofiber mat serves as a tribo-negative film, whereas the PVDF nanofiber mat serves as a tribo-positive film. As demonstrated in step (i), both the layers are isolated from one another and show no charges prior to applying external pressure. When an external force is applied, the upper layer (P-Ar.HBP-3) contacts with the bottom PU layer. Depending on respective electron affinities, equal and opposite charges were formed on contact layers according to the triboelectric principle. PU has a greater tendency to shed electrons than P-Ar.HBP-3. As a result, it becomes positively charged while the adjacent P-Ar.HBP-3 film becomes negatively charged. Electrical balance prevents flow of current in step (ii) because the films gradually separate from one another as you begin to release the external load, which causes a potential difference to develop across the electrodes. In order to equalize the potential differences in step (iii), electrons will migrate from the top to the bottom electrode, and this is repeated until as indicated in step (iv). The electrodes and associated triboelectric films acquire the same amount of contrary charge due to electrostatic induction. As depicted in step (v), if there is an increase in external force, the electrostatic induction balancing becomes disrupted, causing the electrons to flow from bottom to top in the opposite direction. As depicted in Figure 1b(i–v), alternating current is consistently produced by this repetitive cyclical contact and separation mechanism.

Due to their higher triboelectric sensitivity, PVDF fibers with a significant proportion of *β*-phase are more desirable. Therefore, the effect of Ar.HBP-3 upon PVDF crystalline forms was investigated using XRD and FTIR measurements. XRD patterns of electrospun PVDF and P-Ar.HBP-3 (0 to 40 wt.%) fibers are shown in Figure 2a. XRD patterns of neat PVDF can be seen to have two distinct *α*-phase crystallization peaks at 18.3° and 19.8°, which correspond to (020) and (110) reflections, respectively. This outcome demonstrates that the *α*-phase predominates in neat PVDF samples. In the case of P-Ar.HBP-3 (10 wt.%) fiber, unique crystallization peaks corresponding to (020) and (110) reflections of *β*-phase is observed at 20.6°, thereby demonstrating the coexistence of both *α-* and *β*-phases in the blend case. Electrospinning’s simultaneous stretching and poling assists in the PVDF’s transformation from the α- to *β*-phase.

Additional analysis on the crystalline phase changes in P-Ar.HBP-3 (0 to 40 wt.%) was performed using FTIR analyses [91] as shown in Figure 2b. The non-polar *α*-phase is represented by the vibrational bands at 763 and 976 cm^−1^, whereas the electroactive *β*-phase is represented by the distinctive peaks at 841 cm^−1^ and 1276 cm^−1^. The minimal peak intensity of the *γ*-phase at 1234 cm^−1^ indicates the majority presence of *β*- and *α*-phases in their composition. The prominent PVDF fiber’s α-phase peaks sharply reduce when compared to the P-Ar.HBP-3 fiber, indicating that the electrospinning is an effective way to generate and/or improve the polar *β*-phase. This trend is also a confirmation of the effect of Ar.HBP-3 in improving the *β*-phase content in PVDF. Under the condition that the infrared transmittance corresponds with the Lambert–Beer law, the *β*-phase’s relative content in electrospun fibers is determined using the equation:(1)$$F(\beta )=\frac{A_{\beta }}{1.26A_{\alpha }+A_{\beta }}\times 100$$
where 841 cm^−1^ and 763 cm^−1^ absorbance are denoted as *A_β_* and *A_α_*, respectively. Figure 2c depicts how the *β*-phase fractions in electrospun nanofiber have varied. *β*-phase % rises with an increase in Ar.HBP-3 content, reaching a peak value of 93.8 % for the nanofiber, i.e., electrospun PVDF with 10 wt.% Ar.HBP-3 exhibits higher *β*-crystalline phase content than neat PVDF fibers as shown in Figure 2c. The greater interaction between P-Ar.HBP-3 dipoles and the surrounding local electric field aided the significantly increased *β*-phase in PVDF. However, decreasing *β*-phase content with Ar.HBP-3 content beyond 10 wt.% may be attributed to increasing coagulation as observed during sample preparation.

The anticipated mechanism for *β*-phase formation on the P-Ar.HBP-3 surface is shown in Figure 2d. Ar.HBP-3 has a significant role to play towards improving the *β*-phase in PVDF. Similarly, the interfacial interactions of Ar.HBP-3 can pull PVDF link chains to form crystals on the surface in an all-trans form, thereby enabling the conversion of localized non-polar *α*-phase to polar *β*-phase. Ar.HBP-3 serves as a nucleating agent in this circumstance, offering basic units for the creation of PVDF crystalline polymerization and inciting the creation of PVDF’s *β*-phase fragment through potent interaction at the interface [28,56]. In contrast, the interfacial interactions and polarity of –CH_2_/–CF_2_ dipoles differ. As shown in Figure 2d, owing to the strong interaction of hydrogen bonds between F atoms of PVDF and H atoms in hydroxyl groups on the Ar.HBP-3 surface, dipoles of –CH_2_/–CF_2_ in PVDF tend to align their F atoms towards the Ar.HBP-3 at the P-Ar.HBP-3 interface [58]. The mobility and architecture of –CH_2_/–CF_2_ dipoles are affected by these cross-surface links and the dipolar interaction between PVDF and DMF (polar) solvent, thereby causing the formation of the *β*-phase [30,58]. Additionally, during localized poling operations, the conducting Ar.HBP-3 can reinforce the Coulombic force by generating inductive charges and increasing the local electric field. This attracts PVDF chain links to form the *β*-phase at the P-Ar.HBP-3 interface. As a result, when compared to pure PVDF nanofibers, the electrospun P-Ar.HBP-3 nanofiber contains more *β*-phase. Furthermore, it is also observed that when Ar.HBP-3 content is raised further, the *β*-phase proportion is lowered. Increasing the amount of Ar.HBP-3 over 10 wt.% induces agglomeration as well as inhibiting the mobility of PVDF chain links, which may help explain this phenomenon [57]. As a result of Ar.HBP-3 aggregation at the interface, induced charges migrate longitudinally and eventually neutralize the charges in the fiber, which lowers the polarizability of PVDF nanofibers.

Fiber orientation is one of the important factors when processing electrospun fibers, as they are collected on a rotating collector. The two main fiber orientations obtained during electrospinning are ‘aligned’ and ‘random’. While random fibers are randomly oriented at different angles throughout the collected sample, aligned fibers are typically oriented in the same direction. Flat plate and roll-to-roll collectors are the most commonly used collector tools for electrospun nanofibers. The most efficient, and increasingly common method of collecting aligned fibers is by using a rotating drum collector. The degree of fiber alignment is determined by the linear speed of the rotating collector. As linear speed increases, fibers become more aligned. Linear speed on a rotating collector is calculated by υ = r × ω, and is typically represented in revolutions per minute (rpm), where υ = linear velocity, r = radius of drum collector, and ω = angular velocity. In our study using the drum collector, we did not observe any oriented fibers, and instead saw aligned fibers.

SEM images of electrospun PVDF and P-Ar.HBP-3 (10 wt.%) nanofibers are shown in Figure 3a–d, respectively. Furthermore, they display the statistical distributions of the relevant fiber diameters. It is clear that the fibers of both neat PVDF and P-Ar.HBP-3 (10 wt.%) are properly aligned, and neither beads nor non-volatilized solvents can be observed. Ar.HBP-3 (10 wt.%) plays the role of an additive, and showed good structural stability after diffusing into the PVDF fiber. Due to the addition of Ar.HBP-3, the electrospun nanofibers exhibit a smaller diameter than that observed in pure PVDF fibers, which in turn, could improve the electrospinning solution’s electrical conductivity. Under high voltage, the conductive Ar.HBP-3 is easily charged, but as a consequence of the increased Coulombic and electrostatic forces on the Taylor cone, the fiber diameter decreases [71,92].

The elemental mapping images of P-Ar.HBP-3 (10 wt.%) film are shown in Figure 3c–e. Figure 3f–h shows energy-dispersive spectral images of carbon, fluorine, and oxygen elements, respectively. From the SEM data, we were able to study the influence of morphology on the triboelectric properties of the nanofibers.

### 3.2. Electrical Measurements

TENG devices were fabricated using the prepared P-Ar.HBP-3 nanofibers with various Ar.HBP-3 concentrations (0, 10, 20, 30, and 40 wt.%) along with readily available non-woven PU fabric. By applying a constant frequency and force of 1 Hz & 10 N, respectively, the output electrical efficiency of the fabricated TENGs were evaluated. Figure 4a–d depicts the output voltage and current of the PVDF/PU and P-Ar.HBP-3/PU based TENGs. P-Ar.HBP-3 (10 wt.%)/PU-based TENG showed a maximum output voltage and current of 107 V and 1.33 μA, respectively when compared to PVDF/PU which is 12 V and 0.50 μA, respectively. The enhanced *β*-phase and dielectric constant of 10 wt.% P-Ar.HBP-3/PU-based TENG was responsible for the enhanced electrical performance.

Defects surrounding Ar.HBP-3-filled PVDF were eliminated as observed from the cross-sectional FE-SEM images. This results in increasing the PVDF’s *β*-phase and dielectric properties along with a corresponding improvement in the TENG’s electrical output [93]. The electrical response of P-Ar.HBP-3/PU-based TENG under different external mechanical forces at constant frequency is important for its practical and commercial applications. Therefore, the TENG electrical response observed under various applied mechanical forces from 10 to 5 N when the frequency was maintained at 1 Hz exhibited voltages of 35, 49 and 107 V and current of 0.55, 1.14, and 1.33 μA, respectively. The electrical efficiency of the TENG is depicted in Figure 4e,f. It was observed that the TENG’s output voltage and current decreased when the applied force decreased from 10 N to 5 N. From the above observations, P-Ar.HBP-3 (10 wt.%)/PU based TENG was further considered as an optimized device.

### 3.3. Energy Harvesting Applications

To demonstrate the viability of P-Ar.HBP-3/PU-based TENG in real-time applications, the fabricated TENG is able to power 12 LEDs without using a capacitor or rectifier when operating under a load and frequency of 10 N at 1 Hz (Figure 4j,k and Appendix A). The circuit diagram for application studies is shown in Figure 4i. This device generated voltage in a peculiar waveform that could not be used to directly power portable electronics. Therefore, the equivalent voltage produced by the devices was transformed into direct voltage by the use of a full-bridge rectifier circuit while connected to an electronic device that is portable. Furthermore, the rectified voltage was held using a capacitor before connecting any portable electronics. At an operating load of 10 N and a frequency of 1 Hz, the P-Ar.HBP-3/PU-based TENG’s rectified voltage is 46 V, as shown in Figure 4g. For approximately 130 s, the rectified voltage was retained in a variety of capacitors with capacitance values of 2.2, 4.7, 10, and 22 μF storing maximum voltages of 1.4, 1.3, 0.9, and 0.6 V, respectively as shown in Figure 4h.

### 3.4. Wearable Applications

In the ambient environment, mechanical energy is abundant, and the majority is spent in our daily activities. This example illustrates how to use TENGs to collect human motion data. Initially, the sensor is attached to the piezo-amplifier during the measurement, and its output is attached to the Biopac 150 system. This can wirelessly communicate the data to the computer or mobile phone, which uses acknowledgment software during the measurement. The schematic representation of the general circuit used for human health monitoring measurements with Biopac MP150 is shown in Appendix A. P-Ar.HBP-3/PU-based TENG sensor can detect a variety of human body motions including those from wrist, elbow, finger, pocket, mouth and shoe insole. Prior to evaluating a real-time wearable application, the enhanced P-Ar.HBP-3/PU-based TENG (2 × 2 cm^2^) sensitivity is determined (Figure 5a). The sensor is capable of precisely detecting finger tapping, twisting, bending and folding. The sensor produces 3 V at most when it is tapped, 6 V when twisted, 3 V when it is bent, and 2 V when folded.

These results indicate that the TENG responds well enough to twisting compared to other activities as demonstrated in Appendix A. When the sensor is employed as a finger ring with a bending inclination of 45°, it generated 13 V (Figure 5b). As shown in Figure 5c, once the sensor is punched, it exhibits a significant increase in voltage, i.e., 11.8 V. Similar results were seen for the elbow flexion sensed by the TENG’s detecting voltage of 2.5 V, as depicted in Figure 5d. Whenever the sensor is attached to the circular chair, and a person periodically sits on the chair, 16.5 V is displayed (Figure 5e). Additionally, if the sensor is affixed to the exterior of the mask to detect the pattern of coughing, this will vary its signal as depicted in Figure 5f which is 3.4 V and 4.9 V for slow and fast coughing, respectively. As illustrated in Figure 5g–i and in Appendix A, the sensor can differentiate among various human activities including walking, leg movement, and jumping when it is positioned on the pocket, knee flexion and shoe sole. Appendix A shows the real-time measurement for the aforementioned pocket sensor.

As demonstrated in Appendix A, a P-Ar.HBP-3/PU-based TENG sensor can operate as a smart pocket. When the sensor is attached to the pocket and the person in the video puts their hand inside their pocket, it produced voltages of 6.8 V and 5.4 V during walking and hand movements, respectively. In contrast, if the sensor was put on the knee, it produced a voltage of 8.8 V during walking. The sensor first responds well to twisting and folding instead of tapping, and the shoe insole case acts as a tapping mechanism, whereas the other two are bending or folding. Still, the shoe insole generates reduced voltages during walking (2.3 V) and jumping (5.8 V). This implies that the sensor can be used for continuously monitoring the behavior of vehicle drivers and people with paralysis. It is obvious from the results that the P-Ar.HBP-3-based TENG sensors have applications in continuously identifying and distinguishing various physical activities of humans. The above results demonstrate the prepared TENG’s potential for extracting biomechanical energy from the environment of daily living.

## 4. Conclusions

In conclusion, by employing the synergistic enhancing impact of Ar.HBP-3 on the triboelectric efficiency of electrospun PVDF nanofibers, we were able to successfully fabricate flexible and high voltage output TENGs. The polar *β*-phase of PVDF is effectively enhanced using electrospinning technique, aided by the influence of dipole interaction between the functional groups present in PVDF and Ar.HBP-G3. Triboelectric performance of the TENGs is significantly improved in the P-Ar.HBP-3 case compared to neat PVDF. Under the mechanical strain of 10 N at 1 Hz for TENG with 10 wt.% Ar.HBP-3, we could achieve a maximum *V*_OC_ of 107 V and *I*_SC_ of 1.3 μW. Further, to power electronics and charge capacitors, the mechanical energy transformed from triboelectric energy was used. In healthcare monitoring applications, the fabricated TENG exhibited high voltage output stability. In addition, 12 LEDs were directly lit from the maximum *V*_OC_ of 107 V. Additionally, the specified TENG was used to harvest mechanical energy and power portable electronics. In order to track and monitor the patient’s physical conditions, the TENG was also integrated with a variety of health monitoring systems. Due to their low cost, ease of fabrication, and excellent output, the flexible nanofiber TENGs developed in the present study are very promising for powering portable electronics and for healthcare monitoring applications.

## Figures and Tables

**Figure 1 polymers-15-02375-f001:**
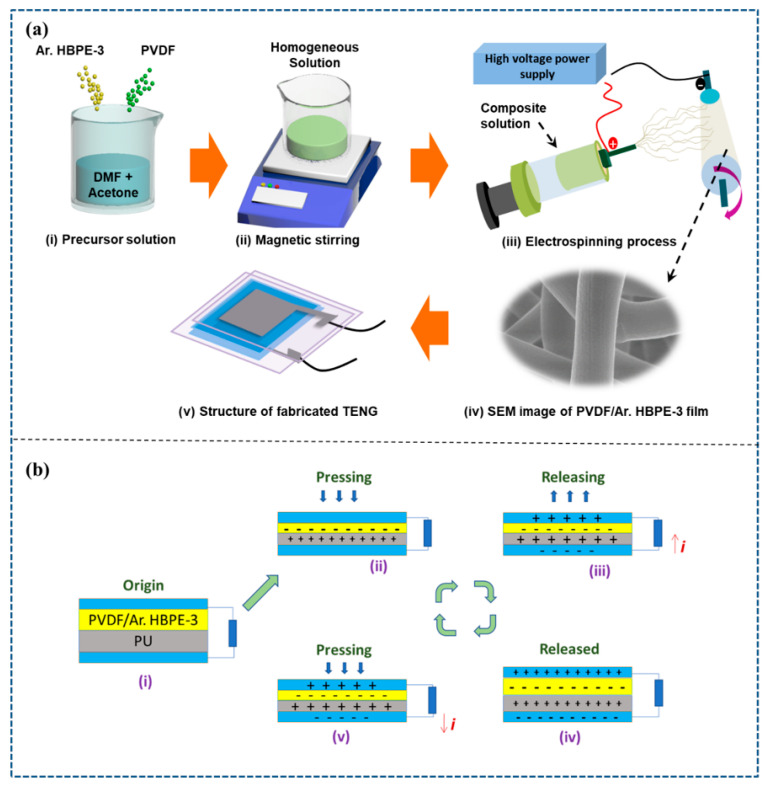
(**a**). Schematic explanation of preparation and electrospinning of P-Ar.HBP-3 solution followed by TENG fabrication: (**i**). preparation of P-Ar.HBP-3 blend solution, (**ii**). solution blending by constant magnetic stirring, (**iii**). electrospinning process, (**iv**). SEM image of electrospun nanofiber and (**v**). fabricated TENG device. (**b**). Schematic illustration of triboelectric mechanism.

**Figure 2 polymers-15-02375-f002:**
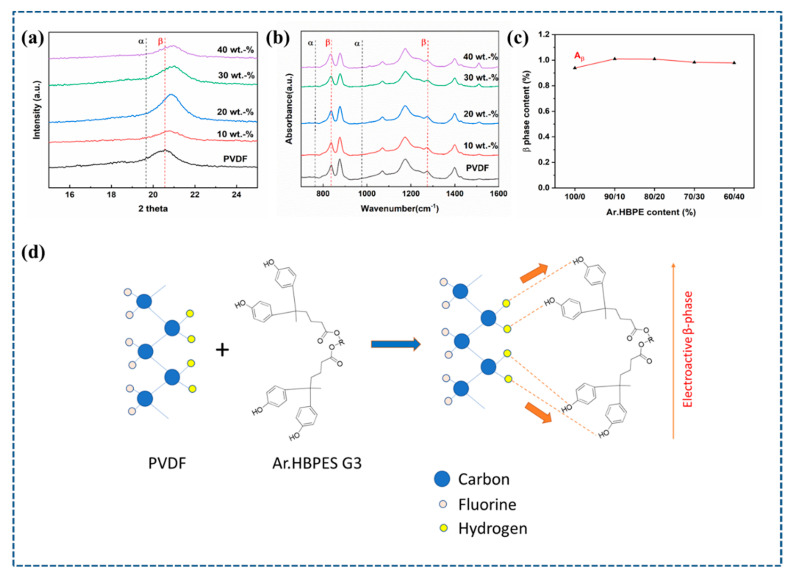
(**a**). XRD pattern and (**b**). FTIR spectra of PVDF fibers as well as P-Ar.HBP-3 (0 to 40 wt.%) blended nanofibers. (**c**). Computed *β*-phase content variability in P-Ar.HBP-3 (0 to 40 wt.%) blended nanofibers, and (**d**). Mechanism of *β*-phase formation in P-Ar.HBP-3 nanofiber.

**Figure 3 polymers-15-02375-f003:**
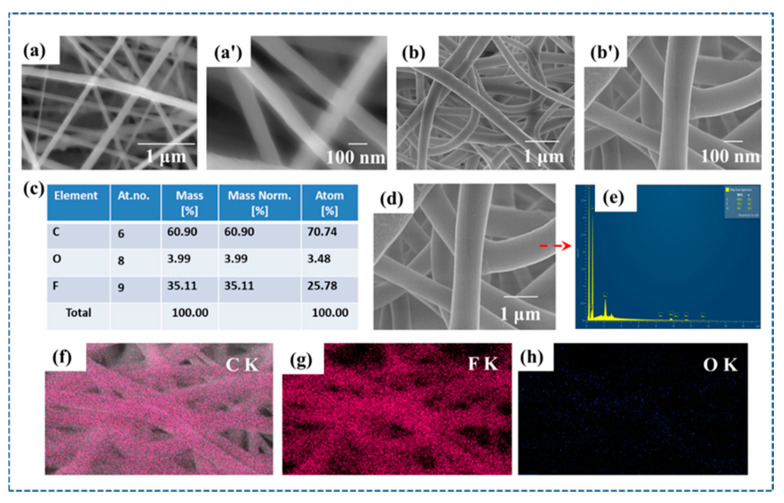
(**a**,**a′**). Surface morphologies of neat PVDF (16 wt.%), (**b**,**b′**). Surface morphologies of P/HBP-3 (10 wt.%) nanofibers, respectively. (**c**,**d**). EDS and elemental mapping of P-Ar.HBP-3 (10 wt.%) nanofiber, respectively. (**e**). Map spectrum of elements in P-Ar.HBP-3 (10 wt.%) nanofiber and (**f**–**h**). Energy-dispersive spectral images of carbon, fluorine and oxygen elements, respectively.

**Figure 4 polymers-15-02375-f004:**
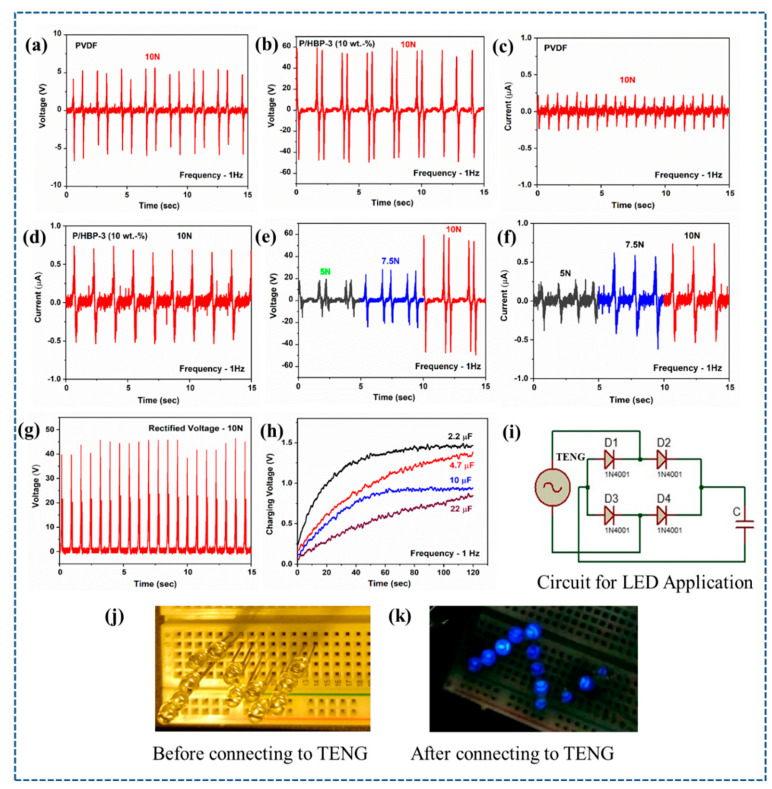
(**a**–**d**). Time-dependent *V*_OC_ and *I*_SC_ graphs of PVDF and P-Ar.HBP-3/PU (10 wt.%)-based TENG under constant frequency of 1 Hz and load of 10 N, respectively. (**e**,**f**). Output performances (*V*_OC_ and *I*_SC_) of P/PU and P-Ar.HBP-3/PU (10 wt.%)-based TENG device as a function of load 5–10 N). (**g**). Rectified voltage of P-Ar.HBP-3/PU (10 wt.%)-based TENG (**h**). P-Ar.HBP-3/PU (10 wt.%)-based TENG device energy storage with various capacitors (2.2–22 μF) with charging cycles. (**i**). Circuit diagram for LED application. (**j**,**k**). Images of 12 LEDs before and after connecting P-Ar.HBP-3/PU (10 wt.%)-based TENG.

**Figure 5 polymers-15-02375-f005:**
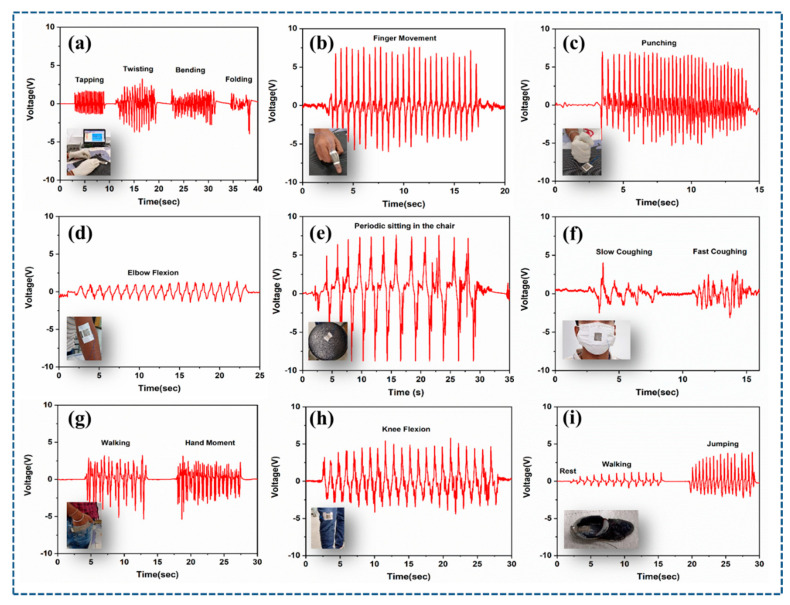
Real-time monitoring of P-Ar.HBP-3/PU (10 wt.%)-based TENG: (**a**). Sensitivity efficiency under various physical deformations of the sensor (tapping, twisting, bending, and folding), (**b**). Sensor’s performance when used as a finger ring, (**c**). Detection of punching intensity, (**d**). Sensitivity of the joint-bending when the sensor is placed at elbow flexion, (**e**). Smart chair health care, (**f**). Detecting and differentiating the intensities of slow and fast coughing pattern while the sensor is used on mouth mask, (**g**–**i**). Motion detection and differentiation (hand moment, leg moment, walk and jump) when the sensor is placed on pocket, knee and shoe insole, respectively for human health motion applications.

## Data Availability

The data will be made available on request.

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
