# Peer review of "Polyvinylidene Fluoride/Aromatic Hyperbranched Polyester of Third-Generation-Based Electrospun Nanofiber as a Self-Powered Triboelectric Nanogenerator for Wearable Energy Harvesting and Health Monitoring Applications"

_polymers, 2023, doi:10.3390/polym15102375_

Round 1
Reviewer 1 Report
Reviewer have some notes to authors:
1) Reviewer recommend to edit the title «Polyvinylidene Fluoride/Aromatic Hyperbranched Polyester of 3rd Generation based Electrospun Nanofiber as Self-powered Triboelectric Nanogenerator for Biomedical Applications» because it not well correlated with obtained results. Please remove the «3rd Generation» and change (or remove) the «Biomedical Applications», because in paper real biomedical application was not demonstrated. With these phrases, the title sounds in hype manner.
2) As usually PVDF is associated with PENG, in this paper authors use TENG approach. Please clarify. What is the origin effect used as application.
3) «HBPE-3, 10 wt.-% sample shows maximum output performances of 107 V which is almost 10 times to the neat PVDF (12 V)». Really? Are they correlated with literature data? Please add in draft comparison with data in literature.
4) Please, add the general circuit for measurements presented on Fig 5.
Author Response
|
Title: Polyvinylidene Fluoride/Aromatic Hyperbranched Polyester of 3rd Generation based Electrospun Nanofiber as Self-powered Triboelectric Nanogenerator for Wearable Energy Harvesting and Health Monitoring Applications |
Author(s): Ramadasu Gunasekhar, Ponnan Sathiyanathan, Mohammad Shamim Reza, Gajula Prasad, Arun Anand Prabu and Hongdoo Kim |
|
Ms. Ref. No: 2283592 |
With reference to the Reviewers thoughts related to certain shortcomings in the manuscript, we have carried out suitable modifications in the revised manuscript to bring more clarity in the discussions, and their justifications are given in detail as follows:
Response to Reviewer-1 Comments
Point 1: Reviewer recommend to edit the title “Polyvinylidene Fluoride/Aromatic Hyperbranched Polyester of 3rd Generation based Electrospun Nanofiber as Self-powered Triboelectric Nanogenerator for Biomedical Applications” because it not well correlated with obtained results. Please remove the 3rd Generation and change (or remove) the Biomedical Applications, because in paper real biomedical application was not demonstrated. With these phrases, the title sounds in hype manner.
Response 1: We agree with the reviewer’s comment on renaming the title. In the revised manuscript, the title is modified as “Polyvinylidene Fluoride/Aromatic Hyperbranched Polyester of 3rd Generation based Electrospun Nanofiber as Self-powered Triboelectric Nanogenerator for Wearable Energy Harvesting and Health Monitoring Applications”.
In case of the word “3rd Generation” in the title, in our lab, we have synthesized 4 different generations (1st, 2nd, 3rd and 4th generation) of hyperbranched polymers (HBP). Among the 4 generations, we are reporting the 3rd generation HBP in this manuscript, and hence the term “3rd Generation” is inlcuded in the title.
Point 2: As usually PVDF is associated with PENG, in this paper authors use TENG approach. Please clarify. What is the origin effect used as application?
Response 2: We agree with the reviewer’s point. In TENG, there is always a pair of triboelectric materials with different charge affinities that are employed to generate charges. Theoretically, to generate more charges or obtain a higher output from the TENG, a significant difference in the charge affinities of the two materials is preferred. However, in practice, the two materials with the highest difference in charge affinity are not selected. The reason is that the tribo-electrification between two materials is based not only on their chemical compositions but also on other physical features, such as the elasticity, friction, and surface topographical structure.
The triboelectric effect and level of contact electrification performance are based on materials that come into contact. In this case, contact electrification is caused by surface transfer of electrons or ions between these two materials. Each material has its own ability to lose or gain electrons during the contact electrification process. This ability can be found in a list of materials (triboelectric series) where polarity and amount of charge for each material are described. An example of the triboelectric series can be seen in Figure 1 based on various works. For example, a material with fluorine functional groups can attract as many electrons as possible from their counterparts. PVDF is a material which tends to gain electrons during electrification. On the other hand, PU is positively charged during electrification. Both materials lie far enough apart in the triboelectric series, and this distance indicates their high triboelectric potential on contact. To support this explanation, we added Refs. 93 and 94, inserted Fig. S1 in SI and also inserted new text in Page 02 in the revised manuscript.
Ref. 93: Tofel, P.; ÄŒástková, K.; Říha, D.; Sobola, D.; Papež, N.; Kaštyl, J.; Ţălu, Åž.; Hadaš, Z. Triboelectric Response of Electrospun Stratified PVDF and PA Structures. Nanomaterials 2022, 12, 1–14, doi:10.3390/nano12030349.
Ref: 94: Yu, A.; Zhu, Y.; Wang, W.; Zhai, J. Progress in Triboelectric Materials: Toward High Performance and Widespread Applications. Adv. Funct. Mater. 2019, 29, 1900098, https://doi.org/10.1002/adfm.201900098.
Figure S1. Triboelectric materials in series following a tendency to easily lose electrons (+) and to easily gain electrons (−).
Point 3: HBPE-3, 10 wt.-% sample shows maximum output performances of 107 V which is almost 10 times to the neat PVDF (12 V). Really? Are they correlated with literature data? Please add in draft comparison with data in literature.
Response 3: We have demonstrated that the triboelectric property of PVDF can be significantly enhanced through the addition of Ar.HBP-3 and aided by electrospinning method. Hence, a high-performance P-HBP-3 (10 wt.-%) TENG has been successfully developed based on this negative triboelectric material. Detailed investigation revealed that the modulation of PVDF composition using HBP-3 and nanofiber structure fabricated through electrospinning contributed significantly to the triboelectric performance enhancement of P-HBP-3 nanofiber. Draft comparison for the above-mentioned voltage is supported by Refs. 6, 13, 23, 54. Also inserted new text in Page 03 in revised manuscript and Table S1 in SI.
Ref. 6: Prasad, G.; Graham, S.A.; Yu, J.S.; Kim, H.; Lee, D.W. Investigated a PLL surface modified Nylon 11 electrospun as a highly tribo-positive frictional layer to enhance output performance of triboelectric nanogenerators and self-powered wearable sensors. Nano Energy 2023, 108, 108178,doi:10.1016/j.nanoen.2023.108178.
Ref. 13: Gajula, P.; Muhammad, F.M.; Reza, M.S.; Jaisankar, S.N.; Kim, K.J.; Kim, H. Fabrication of a silicon elastomer-based self-powered flexible triboelectric sensor for wearable energy harvesting and biomedical applications. ACS Appl. Electron. Mater. 2023, doi:10.1021/acsaelm.2c01773.
Ref. 23: Mekbuntoon, P.; Kaeochana, W.; Prada, T.; Appamato, I.; Harnchana, V. Power output enhancement of natural rubber based triboelectric nanogenerator with cellulose nanofibers and activated carbon. Polymers (Basel). 2022, 14, doi:10.3390/polym14214495.
Ref. 54: Graham, S.A.; Patnam, H.; Manchi, P.; Paranjape, M.V.; Kurakula, A.; Yu, J.S. Biocompatible electrospun fibers-based triboelectric nanogenerators for energy harvesting and healthcare monitoring. Nano Energy 2022, 100, 107455, doi:10.1016/j.nanoen.2022.107455.
|
Reference |
Materials used |
Triboelectric Voltage |
|
6 |
Ny 11 and PNy 11 films against EC5M, EC10M and EC5S, EC10S |
50 to 270 V |
|
13 |
ECF (P-60) against TPU |
65 to 139 V |
|
23 |
NR–CNF–AC |
50 to 130 V |
|
54 |
HPF-F |
50 to 300 V |
Table S1: Draft comparison of triboelectric voltage from the literature data.
Point 4: Please, add the general circuit for measurements presented in Fig 5.
Response 4: We agree with the reviewer’s point. The general circuit for the human health monitoring measurements using Biopac MP150 is represesnted in a schematic way. We also included the same as Figure S3 in SI and mentioned in Page 11 in revised manuscript.
Figure S3: Schematic representation and general circuit for human health monitoring using TENG device.
Overall, the authors thank the reviewers for their valuable comments raised after reviewing this paper. We believe that the comments raised by the reviewers have really helped us to improve the quality of this work. We have sincerely tried to reply for the queries raised by the reviewers and hope that the revised manuscript will not only satisfy the reviewers but also be suitable for publication in Polymers (MDPI).
Thank you
Reviewer 2 Report
· The authors write that electrospun takes place on a rotating collector.
Why authors do not observe oriented fiber on collector.
· If the sensor responds to changes in shape and pressure, have the authors performed abrasion and tearing tests?
· The authors should check the residual solvent in the fabrics produced.
· They should check whether the material is safe in contact with human skin or provide a suitablereference.
Author Response
|
Title: Polyvinylidene Fluoride/Aromatic Hyperbranched Polyester of 3rd Generation based Electrospun Nanofiber as Self-powered Triboelectric Nanogenerator for Wearable Energy Harvesting and Health Monitoring Applications |
Author(s): Ramadasu Gunasekhar, Ponnan Sathiyanathan, Mohammad Shamim Reza, Gajula Prasad, Arun Anand Prabu and Hongdoo Kim |
|
Ms. Ref. No: 2283592 |
With reference to the Reviewers thoughts related to certain shortcomings in the manuscript, we have carried out suitable modifications in the revised manuscript to bring more clarity in the discussions, and their justifications are given in detail as follows:
Response to Reviewer-2 Comments
Point 1: The authors write that electrospun takes place on a rotating collector. Why authors do not observe oriented fiber on collector.
Response: We agree with the reviewer’s point. Fiber orientation is one of the important factors when processing electrospun fibers as they are collected on a rotating collector. The two main fiber orientations obtained during electrospinning are ‘aligned’ and ‘random’. While random fibers are randomly oriented in different angles throughout the collected sample, aligned fibers are typically oriented in the same direction. Flat plate and roll-to-roll collectors are the most used collector tools for electrospun nanofibers. The most efficient, and increasingly common method of collecting aligned fibers is by using a rotating drum collector. The degree of fiber alignment is determined by the linear speed of the rotating collector. As linear speed increases, fibers become more aligned. Linear speed on a rotating collector is calculated by υ = r x ω, and is typically represented in revolutions per minute (rpm), where υ = linear velocity, r = radius of drum collector, and ω = angular velocity. In our study using drum collector, we didn’t observe any oriented fibers, and instead we got aligned fibers. The revised discussion is included in Page 08 in the revised manuscript.
Point 2: If the sensor responds to changes in shape and pressure, have the authors performed abrasion and tearing tests?
Response 2: We agree with the reviewer’s point. The mechanical damage is a big challenge preventing TENGs’ practical applications. TENGs have four working modes, i.e. single electrode mode, contact-separation mode, lateral sliding mode and freestanding triboelectric layer mode. For single electrode and contact-separation modes, TENGs are subjected to constant external tapping, bending, and stretching, which will cause mechanical damages such as cracks, leading to rupture of the sensor structure. In the case of lateral sliding and freestanding triboelectric modes, TENGs suffer from significant friction, wear and adhesion of material during sliding. For ex. the triboelectric surface of TENGs is prone to being worn out by friction forces. The worn particles are transferred to the surface of the opposite material resulting in output degradation and loss in sensor longevity time. Our main focus in this study is on the enhancement of PVDF β-phase by blending with Ar.HBP-3 so as to increase the output voltage of the sensor. Hence, we didn’t perform any mechanical studies for the sensor. Same type of sensor fabrication and studies are already reported Refs. 6, 13, 14, 40 & 54. The revised discussion is included in Page 05, section 3.1. in the revised manuscript.
Point 3: The authors should check the residual solvent in the fabrics produced.
Response 3: We agree with the reviewer’s point. In general, there is very less chance of residual solvent in the electrospun nanoweb produced. During electrospinning process, fast evoporation of the solvents takes place.We also took additional care to remove the traces of residual solvents, if any in the as-spun fibers by keeping the samples in vacuum oven for 5 h at 50 oC to remove any traces of residual solvent present. Futher, we preserved the samples in a desiccator under closed vacuum for a week before proceeding to electrical measurements. Similar procedures were reported for electrospun samples in Refs. 6, 13 & 54. The revised discussion is included in Page 04, section 2.2. in the revised manuscript.
Point 4: They should check whether the material is safe in contact with human skin or provide a suitable reference.
Response 4: We agree with the reviewer’s point. During TENG fabrication process, electrodes attached to nanofiber are encapsulated with PET sheet to avoid short circuit between them. This will not allow direct contact of the skin with sensor, and its safe for doing measurements by directly attaching to human skin. Same type of sensor fabrication is already reported in Refs. 5, 6, 7, 13 & 14. The revised discussion is included in Page 04, section 2.3. in the revised manuscript.
Overall, the authors thank the reviewers for their valuable comments raised after reviewing this paper. We believe that the comments raised by the reviewers have really helped us to improve the quality of this work. We have sincerely tried to reply for the queries raised by the reviewers and hope that the revised manuscript will not only satisfy the reviewers but also be suitable for publication in Polymers (MDPI).
Thank you
Round 2
Reviewer 1 Report
The draft can be published after some grammar editing.
Author Response
Point 1: The draft can be published after some grammar editing.
Response: We agree with the reviewer and we carefully checked our entire manuscript and corrected the typo errors including the grammar editing which are highlighted in the revised manuscript.
Overall, the authors thank the reviewers for their valuable comments raised after reviewing this paper. We believe that the comments raised by the reviewers have really helped us to improve the quality of this work. We have sincerely tried to reply for the queries raised by the reviewers and hope that the revised manuscript will not only satisfy the reviewers but also be suitable for publication in Polymers (MDPI).
Thank you